# UAV Swarms Behavior Modeling Using Tracking Bigraphical Reactive Systems

**DOI:** 10.3390/s21020622

**Published:** 2021-01-17

**Authors:** Piotr Cybulski, Zbigniew Zieliński

**Affiliations:** Faculty of Cybernetics, Military University of Technology, ul. gen. S. Kaliskiego 2, 00-908 Warsaw, Poland; zbigniew.zielinski@wat.edu.pl

**Keywords:** modeling, bigraphs, unmanned aerial vehicles, UAVs, swarm, planning, agent behavior, swarm robotics, multi-agent systems

## Abstract

Recently, there has been a fairly rapid increase in interest in the use of UAV swarms both in civilian and military operations. This is mainly due to relatively low cost, greater flexibility, and increasing efficiency of swarms themselves. However, in order to efficiently operate a swarm of UAVs, it is necessary to address the various autonomous behaviors of its constituent elements, to achieve cooperation and suitability to complex scenarios. In order to do so, a novel method for modeling UAV swarm missions and determining behavior for the swarm elements was developed. The proposed method is based on bigraphs with tracking for modeling different tasks and agents activities related to the UAV swarm mission. The key finding of the study is the algorithm for determining all possible behavior policies for swarm elements achieving the objective of the mission within certain assumptions. The design method is scalable, highly automated, and problem-agnostic, which allows to incorporate it in solving different kinds of swarm tasks. Additionally, it separates the mission modeling stage from behavior determining thus allowing new algorithms to be used in the future. Two simulation case studies are presented to demonstrate how the design process deals with typical aspects of a UAV swarm mission.

## 1. Introduction

Recently, unmanned aerial vehicles (UAVs) have been increasingly used both in the civilian and military spheres, mainly due to their relatively low cost, flexibility, and the elimination of the need for on-board pilot support. The use of UAV swarms is of particular importance, especially with increased autonomy of its elements. It is expected [1] that autonomous UAV swarms will become a key element of future military operations, as well as civilian applications including security, reconnaissance, intrusion detection, and support Search and Rescue (SAR) or Disaster Recovery (DR) operations. DR operations are extremely challenging, and in the immediate aftermath of a disaster, one of the most pressing requirements is for situational awareness. UAV swarms provide an indispensable platform for building the situation awareness in such cases. The obvious benefit of using UAV swarms is an increase in the efficiency of the operation, an accelerated process of its execution and an increased probability of success. Their use in wilderness search and rescue (WiSAR), in particular, has been investigated for fast search-area coverage. One of the most important task in WiSAR is search – until a missing person has been found, they cannot be rescued or recovered. Many search tasks require a number of UAVs to remain in communication at all times and in contact with the base station via a short range ad hoc wireless network. For example, a swarm of UAVs must disperse (take the proper starting positions) to find the missing person as quickly as possible before their energy reserves run out. However, in order to efficiently operate a swarm of UAVs, it is necessary to address the various autonomous behaviors of its constituent elements, sometimes even with conflicting goals, to achieve a high level of adaptation and human-like cognitive behavior. Therefore, it is necessary to conduct research on methods of increasing the autonomy and interoperability of UAVs while limiting global communication and human dependence on the operator.

An individual UAV can perform different tasks, such as terrain reconnaissance, close-up inspection of selected areas, transfer of communication, and target pursuit. The UAV can intelligently take over the role depending on the situation. Such a high degree of adaptation and cooperation in complex scenarios requires innovative solutions at the design stage of the UAV swarm system and appropriate methods of its verification and testing.

A UAV swarm is a special case of robot swarm. There are a few definitions of *robot swarm*, most of them differ in capabilities of its elements and a swarm itself but all of them link it to multirobot systems. A *multirobot system* consists of multiple robots cooperating with a goal to accomplish a given task. The main features associated to multirobot systems are scalability, robustness, flexibility, and decentralized control. In [2], multirobot systems that are not swarms are defined as those that have explicitly stated goals and in which robots are executing individual and/or group tasks. Additionally, robots in such systems have roles that can change during a course of a mission. In the same work, it was pointed out that in swarm system a swarm behavior emerges from local interactions between robots. In [3], the authors defined as a swarm system any robotic system that is capable of performing “swarm behavior”. A frequently quoted definition of robotic swarms is the one presented in [4]:


*“Swarm robotics is the study of how large number of relatively simple physically embodied agents can be designed such that a desired collective behavior emerges from the local interactions among agents and between the agents and the environment.”*


In the same paper, it was recommended for a swarm system to has the following properties.

Robots in a swarm should be autonomous and have the abilities to relocate and interact with other objects in the environment.A swarm control method should allow for coexistence of large number of robots.A swarm can be either homogeneous or heterogeneous. If a swarm is heterogeneous, it consists of multiple homogeneous subgroups.Communication and perceiving capabilities are local. It means that robots do not know a global state of the environment at any moment.

For the purpose of this article, we will adopt the definition presented in [3] with the features described above.

In recent surveys [2,3,5], there are multiple classifications of tasks that a swarm can be given to perform. Below, we will use one that was defined in [5].

Swarm tasks can be divided in three categories:Spatial organization—tasks associated with this category focus on obtaining some spatial property by a swarm. An example of such property might be distance between robots. Typical tasks of this kind are *aggregation*, *dispersion*, *coverage*, and *pattern formation*.Collective motion—this group consists of obstacle avoidance and objects gathering tasks. What makes collective motion different from spatial organization is that in the latter we are mainly focused on rearrangement of individual robots within a swarm while in collective motion we are generally focused on swarm as a whole. Typical tasks in this group are *exploration*, *foraging* (finding and collecting specific objects on the map), *collective navigation* (it aims at constructing, maintaining and, if there is such a need, modifying a formation heading toward some direction ) and *collective transport* (in which a swarm tries to move an object that is otherwise too heavy for a single robot in the swarm).Decision-making—in this group robots make a decision that should lead to a consensus within a swarm. The decision is based on the local perception of the environment and information received from other robots. In the context of robot swarms, this kind of task appears in situations where there is no access to globally shared information. Typical tasks in this group are *consensus* (where a swarm tries to settle on a decision that every of its members agree on), *task allocation* (where robots select from an array of available tasks to perform), and *localization*.

Due to the large number of tasks that a swarm can be given to perform and their complexity, many areas were inspirations for swarm robotics over the course of years. Based on the survey presented in [2], we can distinguish four main areas that served as an inspiration for robot swarms design:Biology—a vast number of solutions and design methods originated directly from observation of real world swarms. To name a few, birds flocks, bees, and ants swarms served as such source of inspiration. A well-known example of robotic swarm originated from biology is presented in [6]. All solutions based on evolutionary processes can be also included in this category. A complex introduction to bioinspired multirobot systems can be found in [7,8].Control theory—this category includes all designs where physical aspects of robots are modeled as continuous-time continuous-space dynamical system and communication between robots is modeled using graph theory. In some works [3], designs based on graph-theory are considered as a separate group. A concise introduction to solutions extensively using control theory can be found in [9]. What is worth noting is that these kinds of designs methods give formal guarantee of correct execution as long as the requirements are met. Unfortunately, this group poorly takes into account indeterministic mission elements and requirements for a swarm are often unrealistic, as it was stated in [10].Amorphous computing and aggregated programming—the main idea behind amorphous computing [11] is to use a large number of identical computers distributed across a space. It is assumed that these computers have only local communication capabilities and do not know their position. Because of its assumptions amorphous computing closely reassembles swarm systems. An example of software implementation of this paradigm is Proto language [12]. In turn, aggregated programming [13] is a paradigm that focuses on the development of large-scale systems from the perspective of their totality rather than individual elements. One prominent aggregate programming approach is based on the field calculus [14]. An implementation of this parading is, for example, Protelis [15]. It is worth noting that it currently used to model IoT-like systems.Physics—swarm design methods inspired by physics are mainly focused on two ideas: artificial forces [16] and Brownian motion [10]. As it was pointed out in [2], a characteristic feature of physics-inspired swarm design methods is that they tend to consider interactions between robots as passive. It means, there may be no message-exchanging communication between agents, instead robots can interact indirectly with each other (most of the time using some kind of forces).

There are multiple taxonomies concerning different aspects of swarm robotics. This includes swarm design methods and methods of analysis of both models and swarms themselves. For example, the taxonomy proposed in [17] distinguishes swarms based on their features, such as their size or communication capabilities. Another taxonomies presented in [18,19] categorize, among others, methods of swarm modeling and its analysis as well as different ways to design its behavior. These taxonomies are especially important for us so we can compare our proposition to the existing methods of swarm design. In [18], the authors divided methods of swarm modeling into two groups. Fist group, called *top-down* (sometimes referred to as macroscopic methods) encapsulate all methods that start from defining a desired swarm behavior and then try to construct robots that exhibit this behavior. The second way to designing robotic swarms, defined as *bottom-up* or microscopic, focuses first on capabilities and behavior of members of a designed swarm. Next, it is checked if the designed swarm is capable of carrying out a given mission. Both design methods have their pros and cons as it was discussed in [20]. The key difference between both methods is where does a design method start from.

In the same work, swarms been distinguished based on their capabilities to improve results. These can be either *non-adaptive*, *learning*, or *evolutionary*. A swarm is *non-adaptive* if the only way to improve its performance is by manual modification by the designer. In turn, a swarm can be described as *learning* if parameters of an algorithm it is using are automatically modified during task execution. Finally, if these parameters are modified in an iterative manner during the design stage with a use of evolution-based techniques, we can describe swarm behavior as *evolving*. In [19], a similar classification of swarm design methods have been proposed. According to this taxonomy, design methods can be described either as *behavior-based* or *automatic*. The first group consists of all methods where a swarm behavior is designed manually by the designer and improved with the trial and error method. The second group is made of all methods where a swarm behavior is constructed without a substantial involvement of the designer.

A constructed swarm model with a behavior policy for the swarm elements can be verified in two ways: using real robots or with simulators. This work is focused on earlier stages of robotic swarm development so we will only briefly cover the key achievements in this field.

The most obvious way to verify a robotic swarm model is with real robots. The most commonly cited swarm robots projects are *swarm-bots* [21], its successor *swarmanoid* [22], and the *Kilobot* project [23]. All of them are capable of performing multiple types of swarm behavior, which suggests that they are all equipped with sufficiently powerful hardware. This, in turn, let us to believe that the lack of common use of robotic swarms is due to insufficient behavior modeling techniques.

Based on an up-to-date state-of-the-art survey [5], it can be seen that there is a number of different simulators designed to help designers verify their work. They vary in terms of performance and versatility of accepted solutions. To name a few, in our opinion two of them are worth to recommend for those wanting to verify their theoretical results:ARGoS [24]—is an open source simulator, whose key features are efficiency, flexibility, and accuracy. According to the information provided by the author, it is used by academic community around the world.CoppeliaSim [25]—(previously known as V-REP) is a very advanced simulator which seems to be used by many commercial and academic institutions globally. It is free for academic use.

One of the proven methods of designing complex systems, which UAV swarm systems certainly are, is engineering based on formal models. Formal models offer a number of possibilities to automate the system design process, including verification of the behavior of the designed system. They allow us to better understand and facilitate analysis of a modeled system. Formal models provide mathematical abstractions of the designed system and can be validated against requirements, tested using various infrastructures, and can also be used to directly simulate the behavior of the system. One of such formalisms which can be used for UAV swarms modeling are bigraphs with tracking. Bigraphs were introduced by R. Milner [26] as a formalism to model systems in which placement and intercommunication between elements play an important role. Despite its novelty, there are already few extensions that allow to broaden its applicability. These are, among others, stochastic bigraphs [27], bigraphs with sharing [28] or bigraphs with tracking [26]. A quick introduction to bigraphs with a real-world use case can be found in [29].

It is important to emphasize that there are currently very few works on robot swarms using bigraphs. Examples [30,31] in the field of multi-agent system do not typically show how to generate behavior policy for swarm elements based on created models. The only solution we have found that does present a method of generating a behavior policy based on bigraphical model was presented in [32]. It uses a basic bigraphical notation mixed with actors model [33]. In our opinion, it is not an automatable method of swarm design.

Currently, there are only few tools supporting design with bigraphs, although it seems there are ongoing works [34] to change that. To our best knowledge, there are only two utilities for designing with bigraphs that are beyond proof-of-concept stage. The most advanced tool for modeling, verifying and simulation of bigraphical system *BigraphER* [35]. The second one, a tool for verification of reachability of states *Bigraphical Model Checker* (*BigMC*) [36] is no longer developed.

In this paper, we will present a method of modeling a UAV swarm with the addition of generating a behavior policy for swarm elements based on constructed model. Our goal is to present a swarm modeling method with the following features:It separates modeling stage from generation of behavior for swarm elements.It is flexible in the meaning that it can be used for a large number of different swarm tasks.It is capable of generating behavior policies on multiple levels of abstractions (from a single agent, through their groups, to an entire swarm as a whole).It is highly automatable. It is a desirable property because it indirectly enforces universal applicability of a method to different design problems. Additionally, automatic methods that are not monolithic tend to be modular, this in turn leads to standardization.

In the next section, we will present a method of modeling UAV swarm systems based on bigraphs with tracking. We will also define a way of constructing behavior policy which will guarantee a successful carrying out a given mission, assuming requirements that had been previously defined are met. Our method is inspired by the work presented in [37]. Although very interesting, it has two major shortcomings. First, the requirement definition stage is loosely coupled with the modeling stage. We wanted to address this issue and allow to formally transform capabilities of robots and mission requirements into model elements. The second issue is the assumption of identical behavior for all swarm elements. We do not consider this a necessary requirement for a swarm, although it may differ depending on the accepted definition of robot swarm.

One of advantages of our method is that the whole process can be automated from the moment of defining mission requirements (as bigraphical patterns) and robots capabilities. We have proved it with software libraries [38,39,40].

To summarize, according to taxonomies presented in [18], our method can be categorized as *bottom-up*, problem-agnostic and a generated behavior can be considered as *non-adaptive*. In turn, using the taxonomies presented in [19], our method can be considered as *automatic* and a method of analysis of a constructed model can be viewed as *macroscopic* (i.e., we are analyzing whole swarm and not individual interactions between its elements).

## 2. Methods and Materials

In this section, we will define formal elements and operations necessary to model a UAV swarm mission and determining a sequence of actions for the swarm elements. We have provided micro-examples at the end of each subsection for easier understanding.

Our proposition can be described as follows. We start from defining a UAV swarm mission as Tracking Bigraphical Reactive System (TBRS). We then transform this TBRS into state space represented as directed multigraph. Finally, we construct a behavior policy for swarm elements. As we treat a state space as a directed multigraph with edges corresponding to actions performed by swarm elements, we can define behavior policy as a walk (a finite length alternating sequence of vertices and edges) from the vertex representing the initial state of the mission to a vertex representing a final state (there can be few of those). A final state is a desirable outcome of the mission. We have proposed a method of finding all walks between any pair of vertices consisting of specified number of edges or loops.

### 2.1. Bigraphs

A bigraph consists of two graphs: a place graph and a link graph. Place graph is intended to model spatial relations between system’s elements. Link graph is a hypergraph that can be used to model interlinking between the elements.

Formally a bigraph is defined as
$$B=(V_{B},E_{B},ctrl_{B},G_{B}^{P},G_{B}^{L}):I\to O$$

$V_{B}$—a set of vertices identifiers;$E_{B}$—a set of hyperedges identifiers;$ctrl_{B}$: $V_{B}\to K$— a function assigning a control type to vertices. *K* denotes a set of control types and is called a signature of the bigraph;$G_{B}^{P}=\langle V_{B},ctrl_{B},prnt_{B}\rangle :m\to n$ and $G_{B}^{L}=\langle V_{B},E_{B},ctrl_{B},link_{B}\rangle :X\to Y$ denote a place and a link graph, respectively. A $prnt_{B}$ function defines hierarchical relations between vertices, roots, and sites. A $link_{B}$ function defines linking between vertices and hyperedges in the link graph;I=〈m,X〉 and O〈n,Y〉 denote an inner face and outer face of the bigraph *B*. By m,n we will denote a set of preceding ordinals of the form: m={0,…,m−1}. Sets *X* and *Y* represent inner and outer names, respectively.

A graphical example of a bigraph is presented in Figure 1.

Reaction rules are used to model dynamics in bigraphical systems. In this paper, we will use simplified tracking reaction rules. We call them simplified because only vertices will be tracked between reactions, as opposed to the original bigraphs with tracking proposed by Milner [26], where both vertices and hyperedges were tracked between reactions. Informally, a reaction rule defines a pattern (redex) in a source bigraph that shall be replaced with another bigraph (reactum). We will omit how patterns are found in bigraphs and how replacement is being done.

Formally, a tracking reaction rule is a quadruple:$$(B_{redex}:m\to I,B_{reactum}:m^{\prime }\to I,\eta ,\tau ),$$
where

$B_{redex}$—redex (a bigraph-pattern to be found in a bigraph to which rule is applied);$B_{reactum}$— reactum ( a bigraph replacing redex );$\eta :m^{\prime }\to m$—a map of sites from reactum to redex;$\tau :V_{reactum}\to V_{redex}$—a partial map of reactum support onto redex support. It allows to indicate which elements are “residues” of a source bigraph in an output bigraph.

An example of reaction rule and its application is presented in Figure 2. A σ function denotes a residue of a source bigraph in an output bigraph.

Having defined the bigraphical reaction rules, we can proceed to the definition of *Bigraphical Reactive System* (BRS). A BRS is a tuple (B,R) where B denotes a set of bigraphs with empty inner face and R is a set of reaction rules defined over B. If R consists of rules with tracking then a pair (B,R) makes a *Tracking Bigraphical Reactive System* (TBRS).

Having a BRS we can generate a Transition System. A *Transition System* is a quadruple: L=(Agt,Lab,Apl,Tra), where

Agt—a set of agents (i.e., bigraphs with an empty inner face, denoted as ϵ);Lab—a set of labels;Apl⊆Agt×Lab—an applicability relation;Tra⊆Apl×Agt—a transition relation;

For the purposes of this work, we will define a *Tracking Transition System* (TTS) $L_{T}=(\mathrm{Agt},\;\mathrm{Lab},\;\mathrm{Apl},\;\mathrm{Par},\;\mathrm{Res},\;\mathrm{Tra})$. First, three elements have the same definition as described above, the rest is defined as follows.

$\mathrm{Par}\left(b,l\right)=p\;b\in \mathrm{Agt},l\in \mathrm{Lab},p\in 2^{V_{b}}$—a participation function. It indicates which elements of a source bigraph participate in a transition. To avoid ambiguity, *Par* function should return an injective mapping between redex’s support of the reaction rule corresponding to the transition’s label and the source bigraph of the transition. We have omitted this in the definitions for the sake of simplicity but the implementation provided in [38] includes this in an output. The definition of the *Par* function provided in this paper allows us to indicate who is participating in a transition but does not indicate what role a participant takes.$\mathrm{Res}\left(b_{1},l,b_{2}\right)=\sigma \left(b_{2}\right)\;b_{1},b_{2}\in \mathrm{Agt},l\in \mathrm{Lab}$—a residue function. It maps vertices in an output bigraph that are residue of a source bigraph to the vertices in the source bigraph;Tra⊆Apl×Agt×Par×Res—a transition relation.

A Tracking Bigraphical Reactive System can be transformed into a Tracking Transition System.

A micro-example of Tracking Transition System is presented in Table 1. Each row describes a single transition in the system. The initial state of the system is presented in the first row in the first *Agt* column. The scenario that this TTS models is as follows. Two UAVs denoted as nodes with controls of type *U* are trying to move from an area of type *A* to an area of type *B*. They can do it in two ways: The first method defined by reaction rule *r1* allows each UAV to move separately. The second method, denoted by reaction rule *r2*, allows both UAVs to move in a cooperative manner. One can think of these reaction rules as of different algorithms enabling various capabilities of the UAVs. We do not provide a graphical representation of reaction rules for this example.

We have prepared a software library for generating Tracking Transition Systems available here [38].

### 2.2. State Space

Having a Tracking Transition System we can transform it into a UAV swarm mission state space. A state space can be later used to generate a behavior for elements of the swarm we can control or have an influence on. Such elements will be called *agents*.

We have taken the following assumptions regarding modeled systems.

The number of agents is constant during whole mission.A system cannot change its state without an explicit action of an agent (alone or in cooperation with other agents).No actions performed by agents is subject to uncertainty.A swarm mission can end for each agent separately in different moment. In other words, agents do not have to finish their part of the mission all at the same time.In case of cooperative actions (actions performed by multiple agents), it is required of all participants to start cooperation at the same moment.

A state space *S* for a system consisting of $n_{a}$ agents and $n_{s}$ states is defined as
S=(N,A,L,I,C,T,M)
where

N⊂N—a set of vertices in the state space. It corresponds to bigraphs in Tracking Transition System;A⊆N×N—a multiset of ordered pairs of vertices. Called set of directed edges;*L*—a set of labels of changes in the system. It will usually consists of reaction rules names from the Tracking Transition System the state space originate from. To determine what changes, in what order, have led to to a specific state we will additionally introduce a set $R=\{l_{t}|l\in L,t\in N\}$.$I=\{N_{1}^{2}\times \cdots \times N_{n_{a}}^{2}\}$—a set of possible state-at-time (SAT) configurations. For example, for $n_{a}=2$ the element $i_{1}=\langle \left(0,777\right),\left(1,123\right)\rangle$ denotes a situation where the agent with id 0 is at the moment 777 while the agent with id 1 is at the moment 123. It is important to emphasize that the configuration $i_{2}=\langle \left(1,123\right),\left(0,777\right)\rangle$ has the same time interpretation but different spatial interpretation. We later show an example with a justification why we need such a set.$C=\left(I\times 2^{R}\right)\cup \left\{0\right\}$—a set of possible mission courses. 0 denotes the neutral element, i.e., $\forall_{x\in C}x+0=0+x=x$, we do not define operation + for the rest of elements of the *C* set.$T=\left\{f_{i}:C\times N\to C\right|i\in N\}\cup \left\{f_{null}\right\}$—a set of functions defining progress of a mission. We later give an example with a rationale why we need such a set. The $f_{null}$ function returns 0 regardless of input. Additionally, we will denote by $T_{i,j}\subset T$ a set of all mission progress functions from the *i* state to the *j* state;M:A→T—a bijection mapping of edges to mission progress functions.

Below, we present an example demonstrating why we needed both *I* and *T* sets.

Let us assume that some TBRS consists of two bigraphs $s_{0}$ and $s_{1}$ as in Figure 3b. Reaction rule for this TBRS is presented in Figure 3a, agents in this system are denoted by the control of type *B*. Then, transform the TBRS into TTS. This TTS consists of two states (associated to both bigraphs) and two transitions (there are two nodes of type *B* and as we can change only one of them there are two ways to do so). Depending on whether the vertex with id 1 or 2 (numbering according to left-hand side of Figure 3b participates in the reaction the result state-at-time configuration will differ. Let us assume that the SAT configuration for the state associated with bigraph $s_{0}$ is equal to $i_{0}=\langle \left(1,0\right),\left(2,0\right)\rangle$ and the reaction with label *r* takes $t_{r}$ units of time. Depending on which vertex participates in the reaction, the SAT configuration for the state $s_{1}$ is either $i_{1}=\langle \left(1,t_{r}\right),\left(2,0\right)\rangle$ or $i_{1}^{\prime }=\langle \left(2,t_{r}\right),\left(1,0\right)\rangle$. Because of this, the corresponding mission progress functions will be of the form $f_{1}\left(\left[\langle (a,x),(b,y)\rangle ,\Omega \right],t\right)=\left[\langle \left(a,x+t_{r}\right),\left(b,y\right)\rangle ,\Omega \cup \left\{l_{t+1}^{2}\right\}\right]$ and $f_{2}\left(\left[\langle (a,x),(b,y)\rangle ,\Omega \right],t\right)=\left[\langle \left(b,y+t_{r}\right),\left(a,x\right)\rangle ,\Omega \cup \left\{l_{t+1}^{1}\right\}\right]$.

Edge identifiers $l^{1}$ and $l^{2}$ denote which way have led to the $s_{1}$ state, their names are arbitrary.

A micro-example of the state space based on TTS from Table 1 is presented in Figure 4 with the mission progress functions defined in Table 2. The key idea behind generating mission progress functions is as follows. For each bigraph *B* (either source or outcome in a transition), we treat a subset of $V_{B}$ (denoting identifiers of agents that we want to determine a behavior policy for) as ordered set. We then compare if the order of agents in the source bigraph has changed in the outcome bigraph. If it did, then we must reflect this change in a tuple being an element of *I* set. In our micro-example, such change of order is particularly visible in the first two transitions (represented by functions $f_{1}$ and $f_{2}$). Both the source and outcome bigraphs of these transitions are the same, yet in the first transition, the order of agents (UAVs) has changed while in the second transition it has not. This is due to the residue function of both transitions. In the first transition, the order of UAVs identifiers (here 1 and 2 in the source bigraph and 1 and 3 in the outcome) are switched, that is, the order in the source bigraph is (1,2) and the order in the outcome bigraph is (3,1). Because of that, the order of the input tuple in $f_{1}$ is changed from 〈(a,x),(b,y)〉 to 〈(b,y+1),(a,x)〉. The incrementation of *y* variable indicates change of time for an agent with identifier equal to value of *b*. In the second transition, the order remained the same, that is, (1,2) (in source) and (1,3) (in the outcome of the transition). The second case for all mission progress functions, returning 0, is necessary to properly define a walk in the state space. Its usage will be explained in the next subsection.

We omit here an algorithm for transformation of TTS into state space but an exemplary implementation of a software doing it is available at [39].

### 2.3. Behavior Policy

We define a behavior policy as a schedule of actions for each agent from the beginning of a mission to its end, without breaks.

Having a state space we can view a behavior policy as a walk indicating what changes and done by who are required in order to reach a desired state.

Before we demonstrate how to construct a proper policy behavior based on a state space, we first need to define the following elements. Please note that by a series we will understand a finite sum of elements.

$K_{s}^{t}=c_{1}+\cdots +c_{m}=\sum_{i=1\cdots m}c_{i}\;c_{i}\in C,s\in \left\{0,\cdots ,n_{s}-1\right\},t\in N$—a series, where summands are mission courses leading to the state *s*;$N_{K}\left(K_{s}^{t}\right)\in N$—a function returning a number of elements in a given series. According to the earlier definition, for any series $K_{s}^{t}$ this function returns a value of *m* (the greatest index of $c_{i}$);$F_{i,j}\left(x,t\right)=\sum_{k\in T_{i,j}}f_{k}\left(x,t\right)\;i,j\in \left\{0,\cdots ,n_{s}-1\right\},t\in N$—a series, where summands are mission progress functions from the *i* to the *j* state;$M_{K}^{t}=\left[\begin{array}{ccc} K_{0}^{t} & \cdots & K_{n_{s}-1}^{t} \end{array}\right],t\in N$—a matrix whose elements are series indicating possible walks leading to each state. Index *t* denotes a number of steps made in a state space. By a step we understand a transition between vertices (including the situation where the initial and final vertex are the same);$M_{F}^{t}=\left[\begin{array}{ccc} F_{0,0}\left(x,t\right) & \cdots & F_{0,n_{s}-1}\left(x,t\right)\\ \cdots & \cdots & \cdots \\ F_{n_{s},0}\left(x,t\right) & \cdots & F_{n_{s}-1,n_{s}-1}\left(x,t\right) \end{array}\right]$—a matrix of transitions between states.

Furthermore, we define two operations:$K_{s}^{t}\circ F_{i,j}\left(x,t\right)=\sum_{k\in T_{i,j}}\sum_{l=1\cdots N_{K}\left(K_{s}^{t}\right)}f_{k}\left(c_{l},t\right)$—a convolution of series defined above;$M_{K}^{t+1}=M_{K}^{t}\cdot M_{F}^{t}$—a multiplication of matrices defined above. Elements of the new matrix are defined by the formula
$$K_{s}^{t+1}=\sum_{k=0}^{n_{s}-1}K_{k}^{t}\circ F_{k,s}\left(x,t\right)$$

With the elements defined above, we can generate all walks consisting of specified number of steps from the initial state to a final state. To do so, one must define the initial state as a $M_{K}^{0}$ matrix and multiply subsequent results by $M_{F}^{i}$ specified number of times. The result will be a $M_{K}^{x}$ matrix, whose elements in the ith column will contain information about all possible walks with *x* steps that ends in the ith state of the state space. If an element in the specified column is equal to 0, then it means there is no such walk.

Going back to our micro-example, using the state space as in Figure 4 with function definitions listed in Table 2 we can determine all sequences of actions that lead to the state denoted as 2nd. Each sequence is equivalent to behavior policy that, when applied, results in moving both UAVs to the area of type *B*.

In order to determine such sequences we create two matrices: a matrix of transitions $M_{F}^{t}$ and matrix of initial state $M_{K}^{0}$. Having both of them, we can multiply subsequent $M_{K}^{t}$ matrices by corresponding $M_{F}^{t}$ matrices and check whether the third state (recall that numbering starts from 0) is reachable. By reachable we understand having value other than 0 in specified column of the $M_{K}^{t}$ matrix.

Definitions of both matrices are listed below.
$$M_{F}^{t}=\left[\begin{array}{ccc} f_{null} & f_{1}+f_{2} & f_{3}\\ f_{null} & f_{null} & f_{4}\\ f_{null} & f_{null} & f_{null} \end{array}\right]$$
$$M_{K}^{0}=\left[\begin{array}{ccc} \left[\langle (1,0),(2,0)\rangle ,\emptyset \right] & 0 & 0 \end{array}\right]$$

The 〈(1,0),(2,0)〉 tuple in the first column of $M_{K}^{0}$ matrix denotes that we have two agents. They are identified as agent 1 and agent 2, although this numbering is arbitrary and could be 777 and 111 as well. The zeros in both (1,0) and (2,0) indicate that both agents starts the mission at the same moment.

Subsequent $M_{K}^{t}$ matrices let us determine how system may change when a specified number of actions occur. For example, $M_{K}^{1}$ gives us information how system may evolve when one action occurs and $M_{K}^{2}$ two actions etc.

In this example, $M_{K}^{1}$ and $M_{K}^{2}$ are of the form
$$M_{K}^{1}=M_{K}^{0}\cdot M_{F}^{0}=\left[\begin{array}{ccc} \left[\langle (1,0),(2,0)\rangle ,\emptyset \right] & 0 & 0 \end{array}\right]\cdot \left[\begin{array}{ccc} f_{null}\left(c,0\right) & f_{1}\left(c,0\right)+f_{2}\left(c,0\right) & f_{3}\left(c,0\right)\\ f_{null}\left(c,0\right) & f_{null}\left(c,0\right) & f_{4}\left(c,0\right)\\ f_{null}\left(c,0\right) & f_{null}\left(c,0\right) & f_{null}\left(c,0\right) \end{array}\right]$$
$$M_{K}^{1}=\left[\begin{array}{ccc} 0 & \left[\langle \left(2,0\right),\left(1,1\right)\rangle ,\left\{r1_{1}\right\}\right]+\left[\langle \left(1,0\right),\left(2,1\right)\rangle ,\left\{r1_{1}\right\}\right] & \left[\langle \left(1,1\right),\left(2,1\right)\rangle ,\left\{r2_{1}\right\}\right] \end{array}\right]$$
$$M_{K}^{2}=M_{K}^{1}\cdot M_{F}^{1}=\left[\begin{array}{ccc} \left[\langle (1,0),(2,0)\rangle ,\emptyset \right] & 0 & 0 \end{array}\right]\cdot \left[\begin{array}{ccc} f_{null}\left(c,1\right) & f_{1}\left(c,0\right)+f_{2}\left(c,1\right) & f_{3}\left(c,1\right)\\ f_{null}\left(c,1\right) & f_{null}\left(c,1\right) & f_{4}\left(c,1\right)\\ f_{null}\left(c,1\right) & f_{null}\left(c,1\right) & f_{null}\left(c,1\right) \end{array}\right]$$
$$M_{K}^{2}=\left[\begin{array}{ccc} 0 & 0 & \left[\langle \left(1,1\right),\left(2,1\right)\rangle ,\left\{r1_{1},r1_{2}\right\}\right]+\left[\langle \left(2,1\right),\left(1,1\right)\rangle ,\left\{r1_{1},r1_{2}\right\}\right] \end{array}\right]$$

We have prepared a software library for generating behavior policies, available here [40].

## 3. Results

The result of our work is a method of design robotic swarms (specifically UAV swarms). In this section, we will present this method with a use of two scenarios related to UAV swarms.

### 3.1. Introductory Scenario

The first of the discussed scenarios aims at presenting how elements defined in Section 2 are meant to be used in modeling a UAV swarm mission. The modeled mission will include the following aspects typical for swarm robotics tasks:CooperationMultiple ways of achieving a goalSynchronization of agents by “idling” while waiting for others

The scenario consists of a map composed of two types of areas and two UAVs of different kind. Let us name the types of areas X and Y while types of UAVs will be called A and B. The initial state of the mission is presented (as a bigraph) in Figure 5. The goal of the mission is to move both UAVs from their initial location to the area of type Y (there is only one such area). In order to demonstrate cooperation, we will restrict possibility of moving from an area of type X to an area of type Y only to simultaneous transition by both UAVs. This can be interpreted in various ways. For example one kind of the UAVs can serve as a navigator when moving between different types of areas, or it can be a carrier that allows the other UAV to travel longer distances. We will not choose one interpretation over the other and rather focus on how to deal with such situations. To additionally increase complexity of the mission we will assume that different kind of UAVs move at different pace. This will enforce a behavior policy to include actions that only purpose is to “kill time” by one UAV while the other, the slower one, will finish their part before engaging in cooperation. It is important to emphasize that we allow agents (UAVs in this example) to idle only by performing actions (either alone or in cooperation) from a fixed set of actions represented by reaction rules.

#### 3.1.1. Bigraphical Reactive System

The TBRS for the first scenario consists of three reaction rules and six bigraphs. The initial state is shown in Figure 5. Controls *A* and *B* represent UAVs of a different kind, while controls *X* and *Y* denote areas of the type with the same name. The reaction rules for the TBRS of this system are presented in Figure 6. The *uta mov atx* reaction, presented in Figure 6a, allows a UAV of type A to move between areas of type X. The *utb mov atx* reaction, depicted in Figure 6b, allows a UAV of type B to do the same. Finally, the *mov atx2y* rule allows for transition of both UAVs between an area of type X to an area of type Y; it is presented in Figure 6c.

The transitions in Tracking Transition System for this scenario are listed in Appendix A.

For simple reference, below is the list of descriptions of each state.

State 0—both UAVs are in the same area of type X that is not at the center of the map. It does not matter which one since all combinations are isomorphic to each other.State 1—the UAV of type A is at the center while the UAV of type B is in an area of type X.State 2—the UAV of type B is at the center while the UAV of type A is in an area of type X.State 3—each of the UAVs is in a distinct area of type X that is not at the center of the map.State 4—both of the UAVs are at the center of the map.State 5—both of the UAVs are in the area of type Y.

#### 3.1.2. State Space

The state space generated from the TTS described above and defined in Appendix A is presented in Figure 7. The set of labels of changes was defined as
$$L=\{m1^{1},m2^{2},m1^{3},m2^{4},m1^{5},m2^{6},m1^{7},m2^{8},m2^{9},m1^{10},m3^{11},m1^{12},m1^{13},m2^{14},m2^{15}\}$$

Mapping of labels to reaction rules was listed in Table 3. For every transition in the TTS there is a corresponding edge in the state space.

As it was stated in the introduction, each type of UAVs moves with different speed. We took the assumption that the UAV of type A needs 1 unit of time to move between areas of type X and the UAV of type B needs 3 units of type to do the same job. Moving between an area of type X to an area of type Y takes 4 units of time (it is done only by two UAVs at once so there is no differentiation by types).

Functions assigned to edges are defined below.
$$f_{1}\left(c,t\right)=\left\{\begin{array}{cc} \left[\langle (a,x+1),(b,y)\rangle ,\Omega \cup \left\{m1_{t+1}^{1}\right\}\right] & :\;c=\left[\langle (a,x),(b,y)\rangle ,\Omega \right]\\ 0 & :\;c=0 \end{array}\right.$$
$$f_{2}\left(c,t\right)=\left\{\begin{array}{cc} \left[\langle (b,y+3),(a,x)\rangle ,\Omega \cup \left\{m2_{t+1}^{2}\right\}\right] & :\;c=\left[\langle (a,x),(b,y)\rangle ,\Omega \right]\\ 0 & :\;c=0 \end{array}\right.$$
$$f_{3}\left(c,t\right)=\left\{\begin{array}{cc} \left[\langle (a,x+1),(b,y)\rangle ,\Omega \cup \left\{m1_{t+1}^{3}\right\}\right] & :\;c=\left[\langle (a,x),(b,y)\rangle ,\Omega \right]\\ 0 & :\;c=0 \end{array}\right.$$
$$f_{4}\left(c,t\right)=\left\{\begin{array}{cc} \left[\langle (b,y+3),(a,x)\rangle ,\Omega \cup \left\{m2_{t+1}^{4}\right\}\right] & :\;c=\left[\langle (a,x),(b,y)\rangle ,\Omega \right]\\ 0 & :\;c=0 \end{array}\right.$$
$$f_{5}\left(c,t\right)=\left\{\begin{array}{cc} \left[\langle (a,x+1),(b,y)\rangle ,\Omega \cup \left\{m1_{t+1}^{5}\right\}\right] & :\;c=\left[\langle (a,x),(b,y)\rangle ,\Omega \right]\\ 0 & :\;c=0 \end{array}\right.$$
$$f_{6}\left(c,t\right)=\left\{\begin{array}{cc} \left[\langle (b,y),(a,x+3)\rangle ,\Omega \cup \left\{m2_{t+1}^{6}\right\}\right] & :\;c=\left[\langle (a,x),(b,y)\rangle ,\Omega \right]\\ 0 & :\;c=0 \end{array}\right.$$
$$f_{7}\left(c,t\right)=\left\{\begin{array}{cc} \left[\langle (a,x),(b,y+1)\rangle ,\Omega \cup \left\{m1_{t+1}^{7}\right\}\right] & :\;c=\left[\langle (a,x),(b,y)\rangle ,\Omega \right]\\ 0 & :\;c=0 \end{array}\right.$$
$$f_{8}\left(c,t\right)=\left\{\begin{array}{cc} \left[\langle (b,y),(a,x+3)\rangle ,\Omega \cup \left\{m2_{t+1}^{8}\right\}\right] & :\;c=\left[\langle (a,x),(b,y)\rangle ,\Omega \right]\\ 0 & :\;c=0 \end{array}\right.$$
$$f_{9}\left(c,t\right)=\left\{\begin{array}{cc} \left[\langle (b,y+3),(a,x)\rangle ,\Omega \cup \left\{m2_{t+1}^{9}\right\}\right] & :\;c=\left[\langle (a,x),(b,y)\rangle ,\Omega \right]\\ 0 & :\;c=0 \end{array}\right.$$
$$f_{10}\left(c,t\right)=\left\{\begin{array}{cc} \left[\langle (a,x+1),(b,y)\rangle ,\Omega \cup \left\{m1_{t+1}^{10}\right\}\right] & :\;c=\left[\langle (a,x),(b,y)\rangle ,\Omega \right]\\ 0 & :\;c=0 \end{array}\right.$$
$$f_{11}\left(c,t\right)=\left\{\begin{array}{cc} \left[\langle (b,z+4),(a,z+4)\rangle ,\Omega \cup \left\{m3_{t+1}^{11}\right\}\right] & :\;c=\left[\langle (a,z),(b,z)\rangle ,\Omega \right]\\ 0 & :\;c\neq \left[\langle (a,z),(b,z)\rangle ,\Omega \right] \end{array}\right.$$
$$f_{12}\left(c,t\right)=\left\{\begin{array}{cc} \left[\langle (a,x),(b,y+1)\rangle ,\Omega \cup \left\{m1_{t+1}^{12}\right\}\right] & :\;c=\left[\langle (a,x),(b,y)\rangle ,\Omega \right]\\ 0 & :\;c=0 \end{array}\right.$$
$$f_{13}\left(c,t\right)=\left\{\begin{array}{cc} \left[\langle (a,x),(b,y+1)\rangle ,\Omega \cup \left\{m1_{t+1}^{13}\right\}\right] & :\;c=\left[\langle (a,x),(b,y)\rangle ,\Omega \right]\\ 0 & :\;c=0 \end{array}\right.$$
$$f_{14}\left(c,t\right)=\left\{\begin{array}{cc} \left[\langle (b,y),(a,x+3)\rangle ,\Omega \cup \left\{m2_{t+1}^{14}\right\}\right] & :\;c=\left[\langle (a,x),(b,y)\rangle ,\Omega \right]\\ 0 & :\;c=0 \end{array}\right.$$
$$f_{15}\left(c,t\right)=\left\{\begin{array}{cc} \left[\langle (b,y),(a,x+3)\rangle ,\Omega \cup \left\{m2_{t+1}^{15}\right\}\right] & :\;c=\left[\langle (a,x),(b,y)\rangle ,\Omega \right]\\ 0 & :\;c=0 \end{array}\right.$$

#### 3.1.3. Behavioral Policy

The initial state for scenario 1 is represented by the vertex with id 0 in Figure 7, the final state is the one with id 5. Our goal is to find a shortest walk from vertex with id 0 to the vertex with id 5; it must not violate the constraint that allow agents to cooperate only when they start in the same moment of time.

Each $T_{i,j}$ set is defined in Table A2 of Appendix B.

Based on *T* set, we can construct $M_{F}^{t}$:$$M_{F}^{t}=\left[\begin{array}{cccccc} f_{null} & f_{1} & f_{2} & f_{null} & f_{null} & f_{null}\\ f_{5} & f_{null} & f_{null} & f_{3} & f_{4} & f_{null}\\ f_{8} & f_{null} & f_{null} & f_{6} & f_{7} & f_{null}\\ f_{null} & f_{10} & f_{9} & f_{null} & f_{null} & f_{null}\\ f_{null} & f_{14}+f_{15} & f_{12}+f_{13} & f_{null} & f_{null} & f_{11}\\ f_{null} & f_{null} & f_{null} & f_{null} & f_{null} & f_{null} \end{array}\right]$$

Knowing that the initial state is associated with vertex with id 0, the $M_{K}^{0}$ is of the form
$$M_{K}^{0}=\left[\begin{array}{cccccc} \left[\langle (1,0),(2,0)\rangle ,\emptyset \right] & 0 & 0 & 0 & 0 & 0 \end{array}\right]$$

By multiplying the above matrices we gain information about changes of the system and how will it affect the agents (UAVs in this case).

For example:$$M_{K}^{1}=M_{K}^{0}\cdot M_{F}^{0}=\left[\begin{array}{cccccc} 0 & \left[\langle (1,1),(2,0)\rangle ,\left\{m1_{1}\right\}\right] & \left[\langle (2,3),(1,0)\rangle ,\left\{m2_{1}\right\}\right] & 0 & 0 & 0 \end{array}\right]$$

In turn, the $M_{K}^{5}$ matrix gives us information about every possible walk leading to the state with id 5 (in general, it gives information about all possible walks consisting of 5 edges). Below are listed all elements of the series being the element of the 6th column (numbering from 0) in the $M_{K}^{5}$ matrix:$\left[\langle (1,7),(2,7)\rangle ,\left\{m3_{5}^{11},m2_{4}^{4},m1_{3}^{1},m1_{2}^{5},m1_{1}^{1}\right\}\right]$$\left[\langle (1,7),(2,7)\rangle ,\left\{m3_{5}^{11},m2_{4}^{4},m1_{3}^{1},m1_{2}^{3},m1_{1}^{1}\right\}\right]$$\left[\langle (1,7),(2,7)\rangle ,\left\{m3_{5}^{11},m1_{4}^{7},m2_{3}^{2},m1_{2}^{5},m1_{1}^{1}\right\}\right]$$\left[\langle (1,7),(2,7)\rangle ,\left\{m3_{5}^{11},m1_{4}^{7},m2_{3}^{9},m1_{2}^{3},m1_{1}^{1}\right\}\right]$$\left[\langle (1,7),(2,7)\rangle ,\left\{m3_{5}^{11},m2_{4}^{7},m1_{3}^{13},m2_{2}^{4},m1_{1}^{1}\right\}\right]$$\left[\langle (1,7),(2,7)\rangle ,\left\{m3_{5}^{11},m1_{4}^{7},m1_{3}^{13},m1_{2}^{7},m2_{1}^{2}\right\}\right]$$\left[\langle (1,7),(2,7)\rangle ,\left\{m3_{5}^{11},m1_{4}^{7},m1_{3}^{12},m2_{2}^{4},m1_{1}^{1}\right\}\right]$$\left[\langle (1,7),(2,7)\rangle ,\left\{m3_{5}^{11},m1_{4}^{7},m1_{3}^{12},m1_{2}^{7},m2_{1}^{2}\right\}\right]$

Using the above we can define a behavior policy (i.e., a schedule of actions) for each UAV.

For example using the first element of the 6th column we get the following walk.
$$0\overset{m1_{1}^{1}}{\to }1\overset{m1_{2}^{5}}{\to }0\overset{m1_{3}^{1}}{\to }1\overset{m2_{4}^{4}}{\to }4\overset{m3_{5}^{11}}{\to }5$$

This can be further transformed into a behavior policy as presented in Table 4.

### 3.2. More Advanced Example

The second example is intended to present a more realistic UAV swarm mission. Additionally, we present two propositions of metrics for measuring size of swarm robotic systems.

In this scenario, the goal is to collect all information located on a map and secure it in a base. We have made the following assumptions, in regard to the mission.

Every UAV is capable of storing and transporting up to one information at the time.All sources of information can transmit information to any number of UAVs in parallel.An information can be secured in a base only when the UAV containing the information is inside the base.Any number of UAVs can secure information in a base at the same time.

The above mission will be resolved in four different variants of the initial state:The map consists of two areas, one UAV, one source of information, and one information.The map consists of four areas, two UAVs, two sources of information, and two information.The map consists of four areas, two UAVs, two sources of information, and four information.The map consists of nine areas, three UAVs, two sources of information, and four information.

#### 3.2.1. Bigraphical Reactive System

Regardless of the variant, the BRS for the second scenario consists of six reaction rules, presented in Figure 8, Figure 9, Figure 10, Figure 11, Figure 12 and Figure 13. All of the initial states for different variants of the scenario are presented in Figure 14. All site mapping functions and residue functions are identities. Table 5 lists all control types with respective real world objects they are representing.

The *move* reaction rule:
Figure 8Reaction rule *move* for scenario 2.
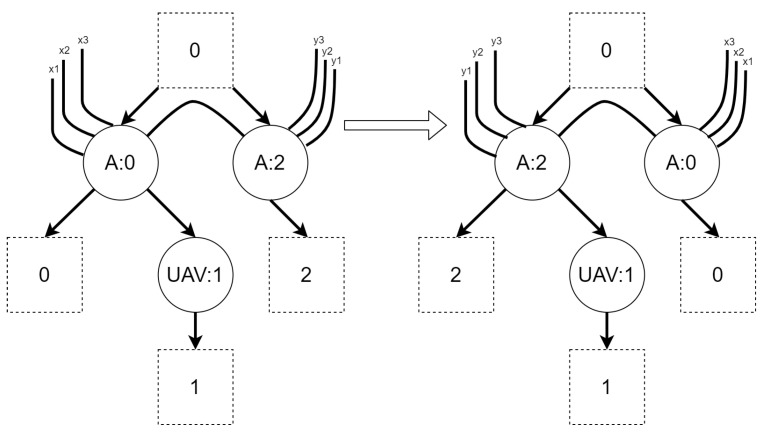
Rationale: the rule is intended to allow a UAV to move between areas.The *move into base* reaction rule:Figure 9Reaction rule *move into base* for scenario 2.
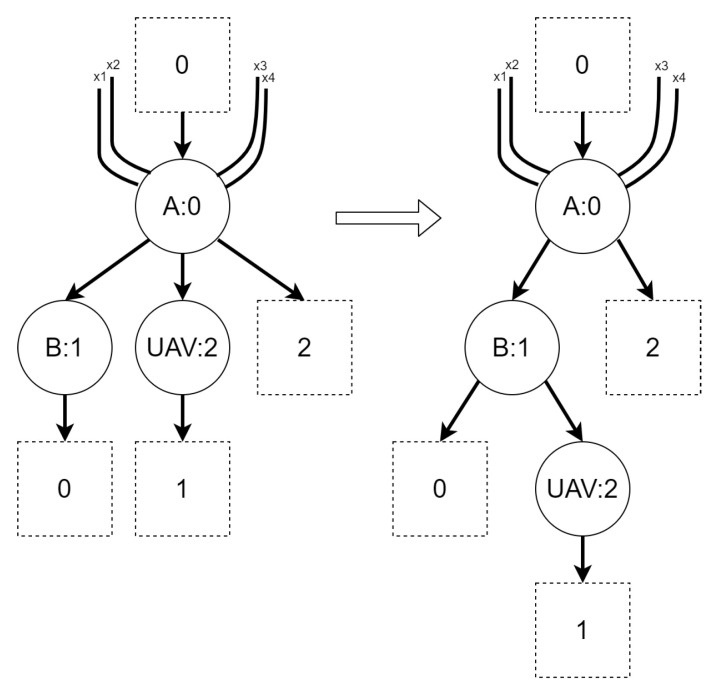
Rationale: the rule is intended to allow a UAV to move into a base.The *move out of base* reaction rule:Figure 10Reaction rule *move out of base* for scenario 2.
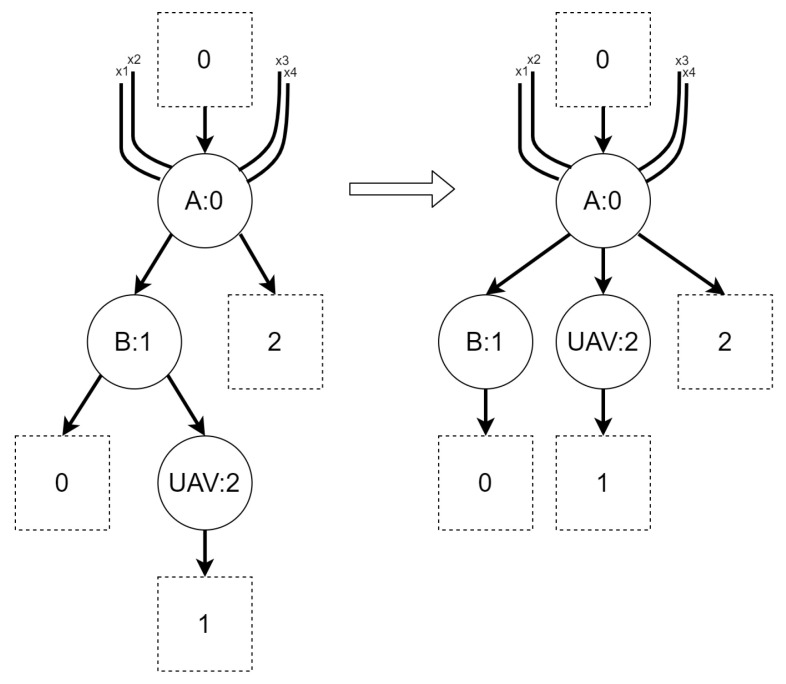
Rationale: the rule allows to move a UAV out of base.The *download data* reaction rule:Figure 11Reaction rule *download data* for scenario 2.
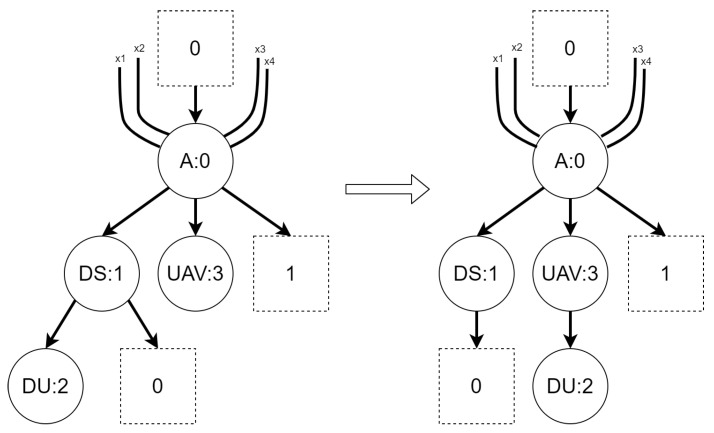
Rationale: the rule is intended to allow a UAV to download information from an information source.The *deploy data* reaction rule:Figure 12Reaction rule *deploy data* for scenario 2.
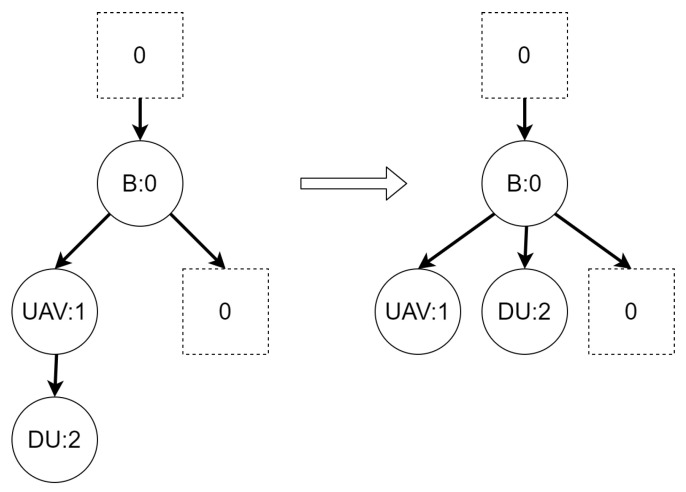
Rationale: the rule is intended to allow UAVs to secure information in a base.The *deactivate uav* reaction rule:Figure 13Reaction rule *deactivate uav* for scenario 2.
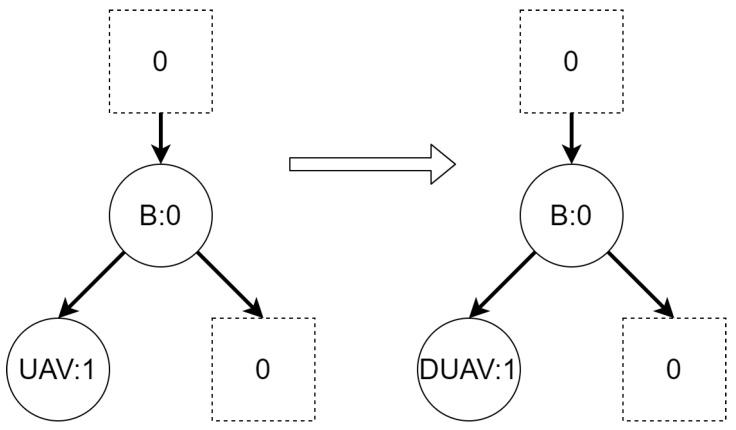
Rationale: the rule is intended to allow a UAV to get deactivated. We will consider the mission be finished when all information are secured and all UAVs are deactivated.

Initial states of the system in each variant are presented in Figure 14a–d. The numbering corresponds to the sequence of the variants descriptions above.

**Figure 14 sensors-21-00622-f014:**
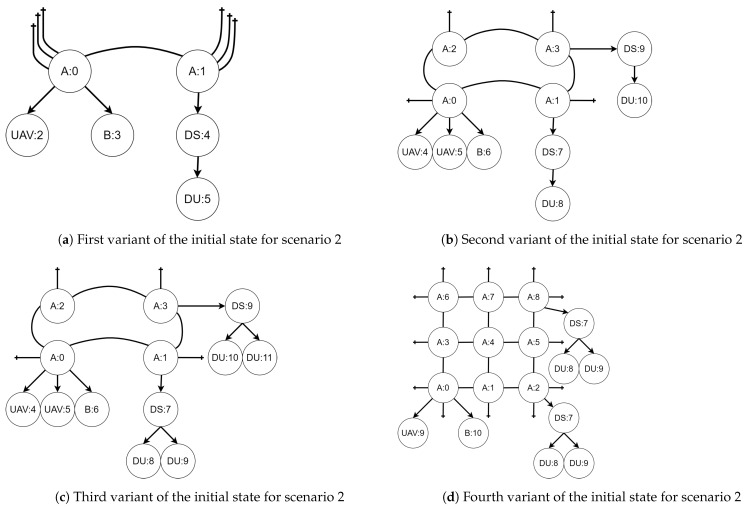
Initial states of scenario 2 in all variants.

#### 3.2.2. State Space

From the above reactive systems (each generated from different initial state), we can generate four state spaces. Due to their large size (see Figure 15), their details, such as graphical representation and transition matrices, will be defined only for the first (smallest) variant.

A relation between variant of scenario 2 and its size is depicted in Figure 15. The state space generated from TTS of the first variant of scenario 2 is shown in Figure 16.

Mission progress functions definitions are listed below.
$$f_{1}\left(\left[\langle (a,x)\rangle ,\Omega \right],t\right)=\left[\langle (a,x+1)\rangle ,\Omega \cup \left\{mib_{t+1}\right\}\right]$$
$$f_{2}\left(\left[\langle (a,x)\rangle ,\Omega \right],t\right)=\left[\langle (a,x+1)\rangle ,\Omega \cup \left\{mov_{t+1}\right\}\right]$$
$$f_{3}\left(\left[\langle (a,x)\rangle ,\Omega \right],t\right)=\left[\langle (a,x+1)\rangle ,\Omega \cup \left\{dea_{t+1}\right\}\right]$$
$$f_{4}\left(\left[\langle (a,x)\rangle ,\Omega \right],t\right)=\left[\langle (a,x+1)\rangle ,\Omega \cup \left\{mob_{t+1}\right\}\right]$$
$$f_{5}\left(\left[\langle (a,x)\rangle ,\Omega \right],t\right)=\left[\langle (a,x+1)\rangle ,\Omega \cup \left\{dod_{t+1}\right\}\right]$$
$$f_{6}\left(\left[\langle (a,x)\rangle ,\Omega \right],t\right)=\left[\langle (a,x+1)\rangle ,\Omega \cup \left\{mov_{t+1}\right\}\right]$$
$$f_{7}\left(\left[\langle (a,x)\rangle ,\Omega \right],t\right)=\left[\langle (a,x+1)\rangle ,\Omega \cup \left\{mov_{t+1}\right\}\right]$$
$$f_{8}\left(\left[\langle (a,x)\rangle ,\Omega \right],t\right)=\left[\langle (a,x+1)\rangle ,\Omega \cup \left\{mib_{t+1}\right\}\right]$$
$$f_{9}\left(\left[\langle (a,x)\rangle ,\Omega \right],t\right)=\left[\langle (a,x+1)\rangle ,\Omega \cup \left\{mov_{t+1}\right\}\right]$$
$$f_{10}\left(\left[\langle (a,x)\rangle ,\Omega \right],t\right)=\left[\langle (a,x+1)\rangle ,\Omega \cup \left\{ded_{t+1}\right\}\right]$$
$$f_{11}\left(\left[\langle (a,x)\rangle ,\Omega \right],t\right)=\left[\langle (a,x+1)\rangle ,\Omega \cup \left\{mob_{t+1}\right\}\right]$$
$$f_{12}\left(\left[\langle (a,x)\rangle ,\Omega \right],t\right)=\left[\langle (a,x+1)\rangle ,\Omega \cup \left\{dea_{t+1}\right\}\right]$$
$$f_{13}\left(\left[\langle (a,x)\rangle ,\Omega \right],t\right)=\left[\langle (a,x+1)\rangle ,\Omega \cup \left\{mob_{t+1}\right\}\right]$$
$$f_{14}\left(\left[\langle (a,x)\rangle ,\Omega \right],t\right)=\left[\langle (a,x+1)\rangle ,\Omega \cup \left\{mov_{t+1}\right\}\right]$$
$$f_{15}\left(\left[\langle (a,x)\rangle ,\Omega \right],t\right)=\left[\langle (a,x+1)\rangle ,\Omega \cup \left\{mib_{t+1}\right\}\right]$$
$$f_{16}\left(\left[\langle (a,x)\rangle ,\Omega \right],t\right)=\left[\langle (a,x+1)\rangle ,\Omega \cup \left\{mov_{t+1}\right\}\right]$$

Table 6 shows the mapping of labels of changes in the system to reaction rules that led to the changes between states.

#### 3.2.3. Behavior Policy

Similarly as in Section 3.2.2, we will limit ourselves to the first variant of scenario 2.

The $T_{i,j}$ sets for the first variant of scenario 2 are listed in Table A3 in Appendix B.

Assuming, the initial state is represented by the vertex with id 0, the $M_{K}^{0}$ matrix is of the form
$$M_{K}^{0}=\left[\begin{array}{cccccccccc} \left[\langle (1,0)\rangle ,\emptyset \right]0 & 0 & 0 & 0 & 0 & 0 & 0 & 0 & 0 & 0 \end{array}\right]$$

We will omit the definition of $M_{F}^{t}$, hoping that its form is obvious knowing the sets presented in Appendix B.

If the final state is represented by the vertex with id 8, then the first walk will be found in $M_{K}^{6}$ matrix and it will be of the form
$$0\overset{mov_{1}}{\to }2\overset{dod_{2}}{\to }4\overset{mov_{3}}{\to }5\overset{mib_{4}}{\to }6\overset{ded_{5}}{\to }7\overset{dea_{6}}{\to }8$$

## 4. Discussion

In this paper, we have presented a method of modeling a UAV swarm mission using bigraphs with tracking as well as a method of generating a behavior policy for elements of the swarm. The proposed method has the desired properties described in the introduction. One of the main advantages of the proposed method is the possibility to fully automate the process of determining the behavior of swarm elements from the phase of defining mission requirements (as bigraphical patterns) and UAV capabilities. The proposed method is flexible in terms of using it for different swarm tasks; it is also modular and capable of generating behavior policies on multiple levels of abstractions. Because of its modularity, we can modify some of the method’s modules while leaving the rest unchanged. For example, there may be a need of defining a function to evaluate walks generated in the last stage of a design process. It is possible without modifying neither the way the mission is modeled (the previous step) nor how the schedule of actions is constructed based on the result walk (the next step). The method was verified on two scenarios. Additionally, a software to verify the calculated results has been developed.

One of the conclusions from the work is the observation of how quickly a system’s size is increasing. In case of a real-world scenario, we can safely assume a size of the system to be in the order of millions of states and dozens of millions of transitions. Based on our tests, current software is capable to effectively support a designer in a process of modeling a system consisting of up to dozens of thousands of states. Because of this, it is reasonable to point out that in order to use our method for a real world use case it is needed to develop more efficient implementations of operations on bigraphs and operations defined in this article. To the best of our knowledge, none of the existing methods of design of UAV swarms are universal, they are either suitable for a specific task or a single group of them (such as *spatial organization* or *decision-making*) at best. Despite classifying our method as problem-agnostic we are aware that assuming that system’s change can only be triggered by its agents and non-adaptiveness of behavior policy are quite restrictive. This is a limitation of its applicability to tasks from *spatial organization* or *collective motion* category. Some of the tasks in the latter category may regard situations where environment can significantly change regardless of UAVs actions, an example of this may be a search and rescue operation considered as a special case of foraging task. In such cases, our method is not suitable in its current form. Summarizing, we can recommend our approach to tasks in *spatial organization* category in general and a subset of *collective motion* tasks. *Decision-making* tasks are generally unsuitable to be solved by our method because the behavior policy it generates is non-adaptive and most of the decisions in this category concern events occurring at hand.

Currently, we are unable to quantitatively compare our method to any alternatives. The key factor that may determine whether to choose our approach over others may be the need of reusability. The method presented in this paper generates a behavior policy for a specific task and is obviously inferior in situations where there is a need for generic behavior that is adjustable with some parameters.

We can distinguish the following goals for future work.

Developing a method of policy behavior generation that takes into account agents that are beyond control of the designer. An example of such agent might be a person.Developing a method of behavior policy generation that considers indeterministic nature of agents’ actions.Developing a method of behavior policy generation for missions with a variable number of agents over their course.More efficient software providing the functionality described in this paper. The current version of the software should be considered as proof-of-concept.

## Figures and Tables

**Figure 1 sensors-21-00622-f001:**
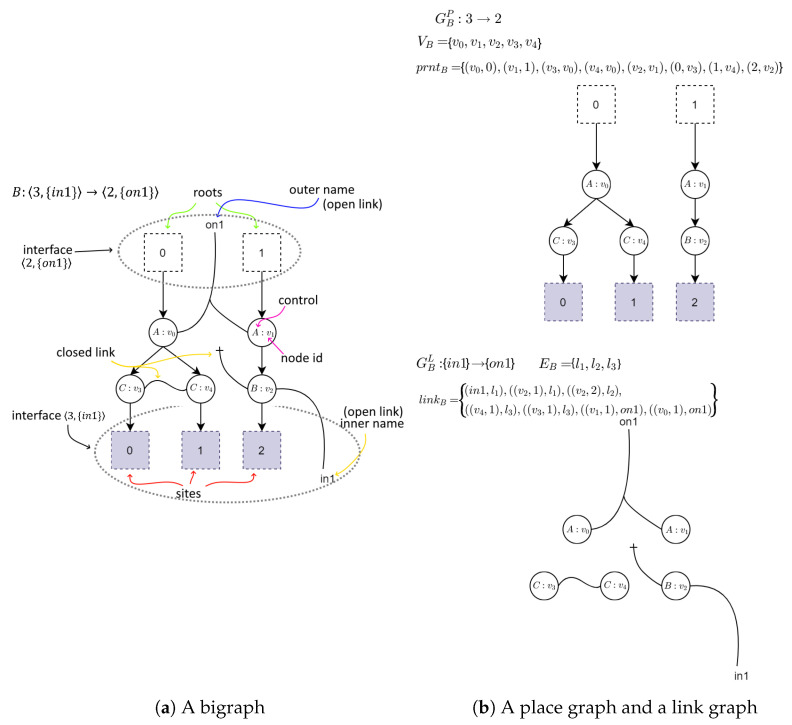
An example of a bigraph and its constituents. The right part represents a place graph (the upper part of the figure) and a link graph (the lower part of the figure). They share a signature which defines control types (letters in nodes) and arity of each control (number of unique links that can be connected to a node with specified control). On the left there is the bigraph made from the superposition of them both.

**Figure 2 sensors-21-00622-f002:**
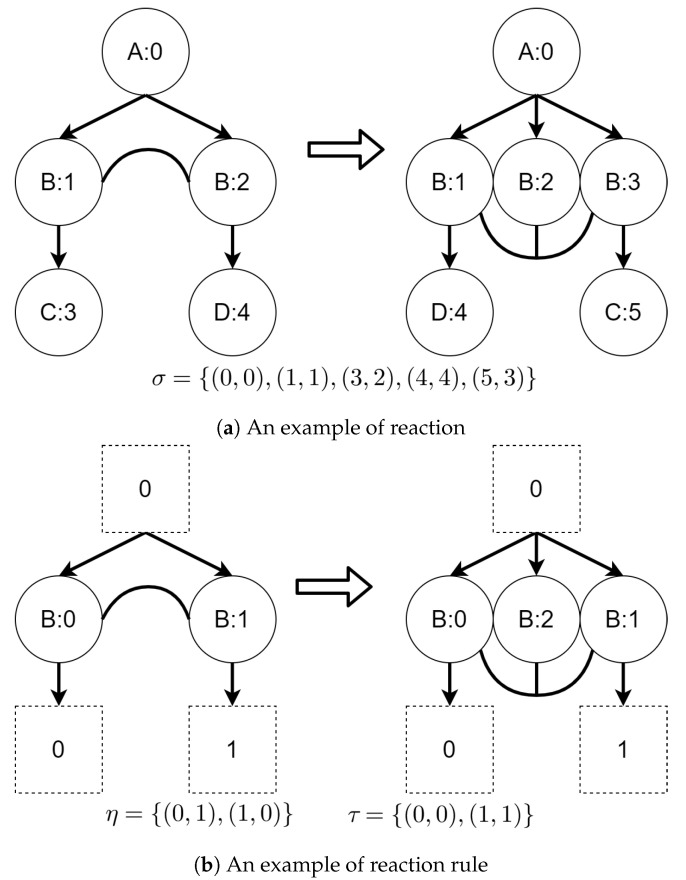
An example of bigraphical reaction with a corresponding reaction rule. The switch of vertices with controls *C* and *D* is caused by the *η* function. The *σ* mapping denotes which vertex in the output bigraph corresponds to which vertex in the source bigraph. It shows that the vertex with id 2 is “new” (it is not a residue of the source bigraph).

**Figure 3 sensors-21-00622-f003:**
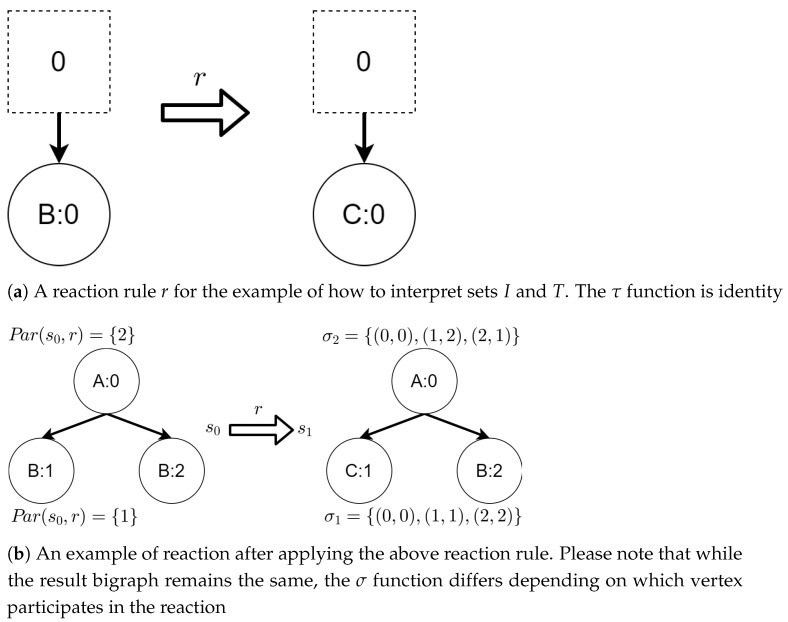
A TTS for the example of how to interpret sets *I* and *T*.

**Figure 4 sensors-21-00622-f004:**
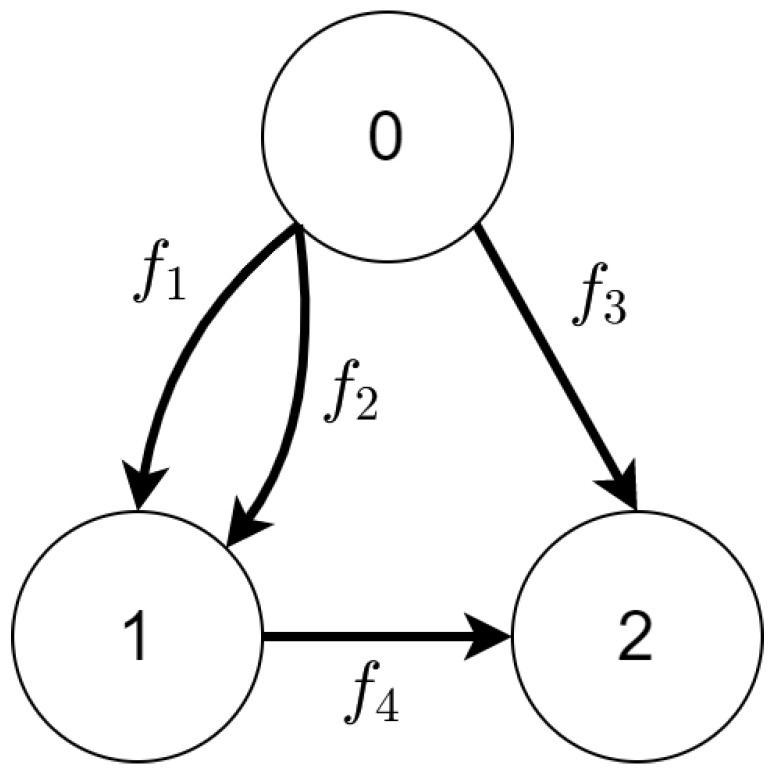
The state space generated from Tracking Transition System defined in Table 1. Mission progress functions definitions are defined in Table 2.

**Figure 5 sensors-21-00622-f005:**
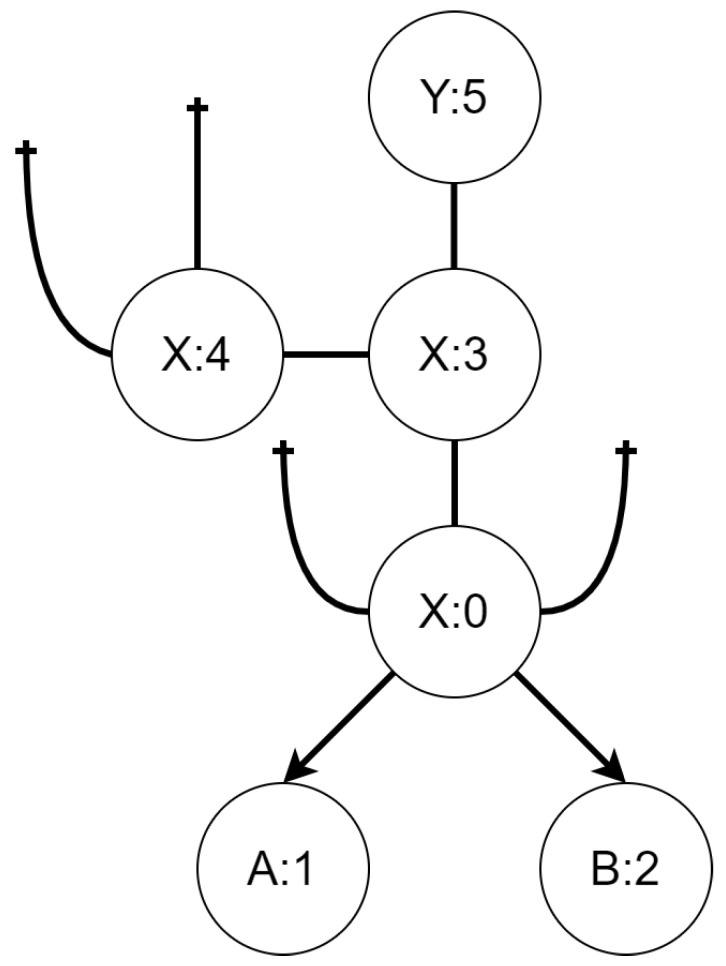
Initial state of the system modeled in scenario 1. X and Y are different types of areas, while A and B are different kinds of UAVs.

**Figure 6 sensors-21-00622-f006:**
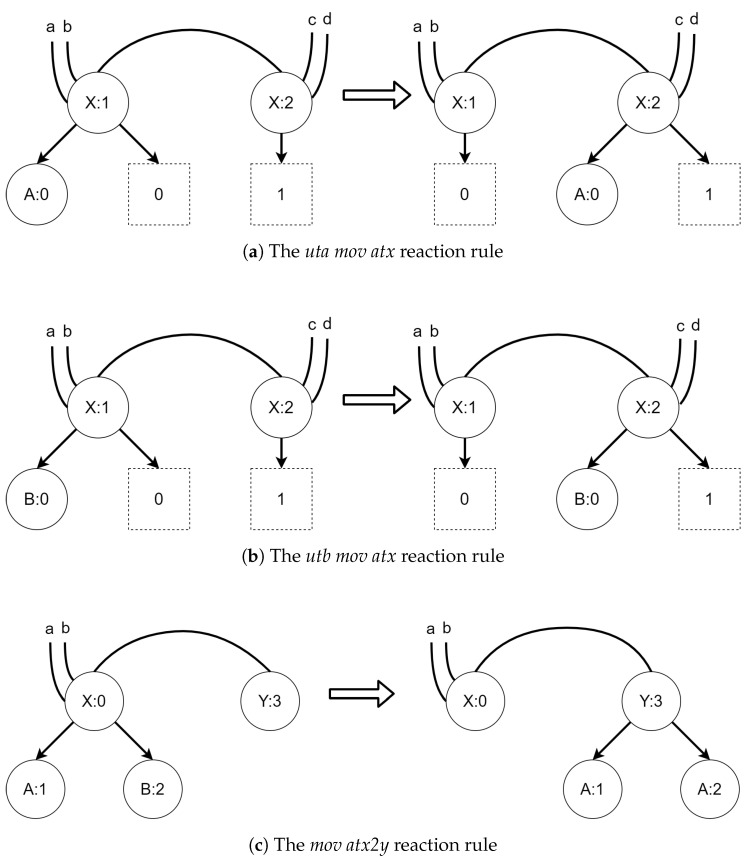
The reaction rules for the first scenario. All of the site mappings and residue functions are identities.

**Figure 7 sensors-21-00622-f007:**
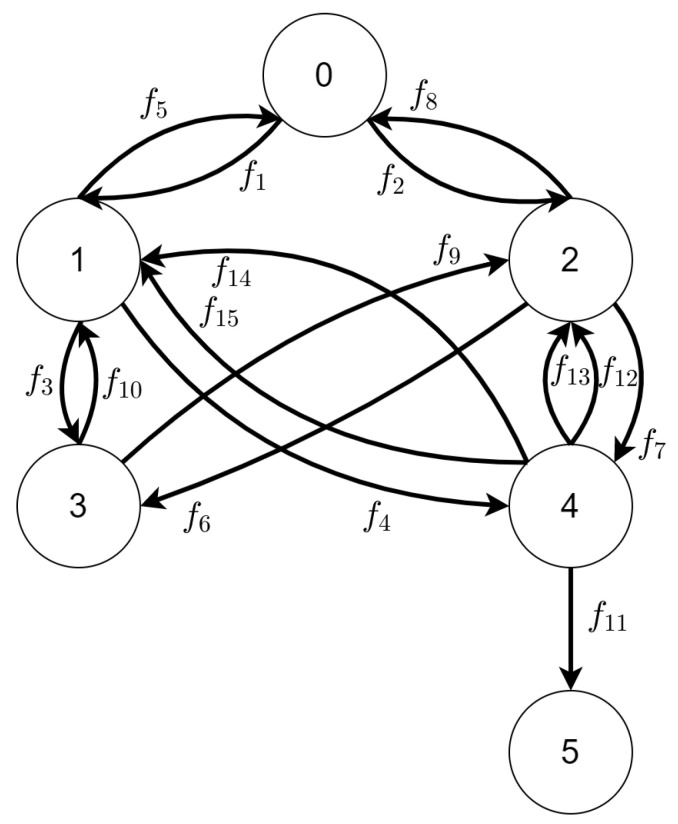
The state space for scenario 1 based on Tracking Transition System defined in Appendix A.

**Figure 15 sensors-21-00622-f015:**
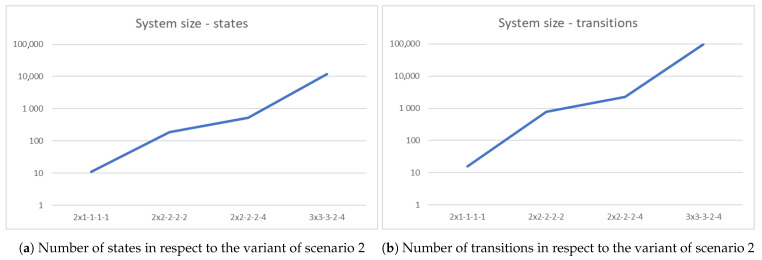
Size of scenario 2 system for all its variants.

**Figure 16 sensors-21-00622-f016:**
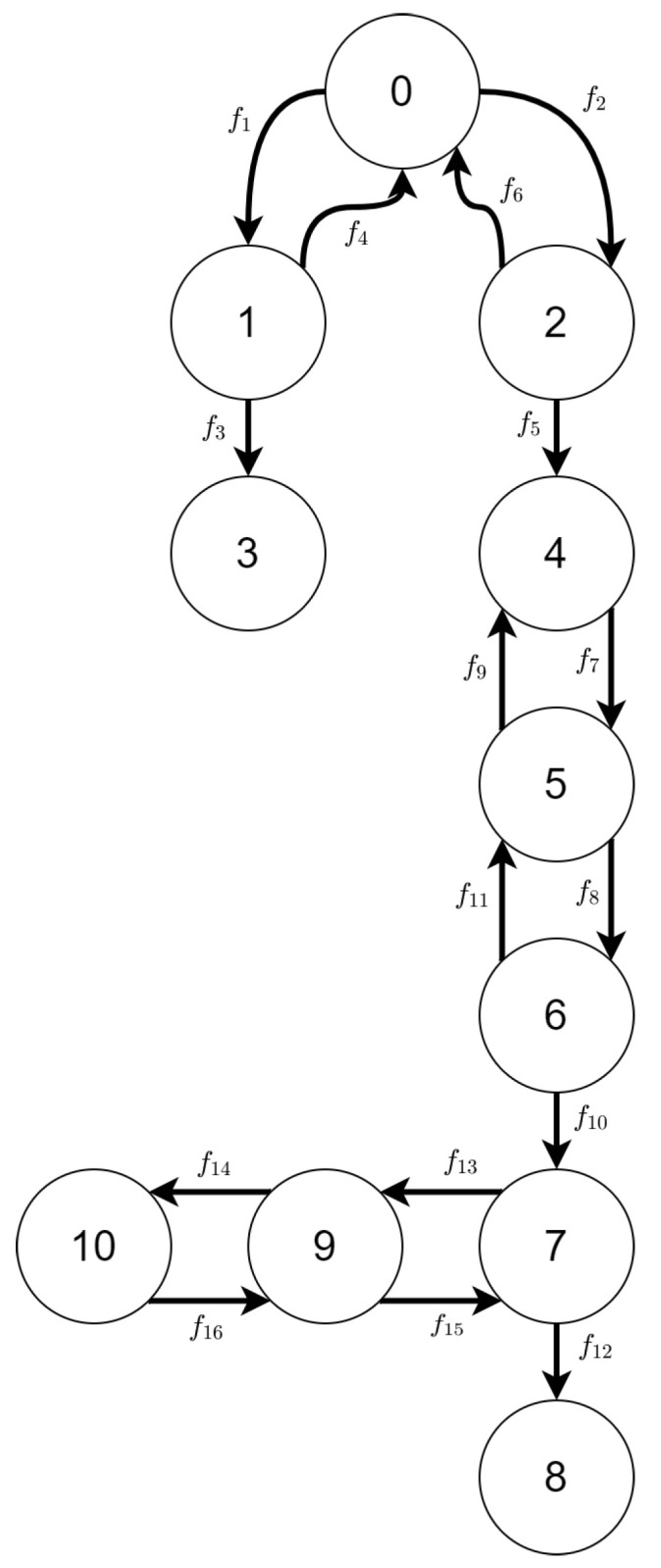
The state space of the first variant of scenario 2.

**Table 1 sensors-21-00622-t001:** An example of a Tracking Transition System. Each row defines a single transition in the system. The initial state is defined in the first column of the first row. The definition of two reaction rules used to generate this TTS were omitted but they allow to move either one or two nodes of type *U* from *A* to *B* at once.

Agt	Lab	Agt	Par	Res
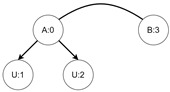	*r1*	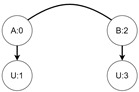	Par={0,1,3}	σ={(0,0),(1,2),(2,3),(3,1)}
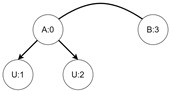	*r1*	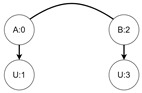	Par={0,2,3}	σ={(0,0),(1,1),(2,3),(3,2)}
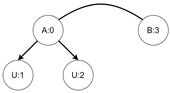	*r2*	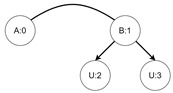	Par={0,1,2,3}	σ={(0,0),(1,3),(2,1),(3,2)}
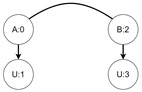	*r1*	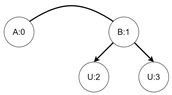	Par={0,1,2}	σ={(0,0),(1,2),(2,3),(3,1)}

**Table 2 sensors-21-00622-t002:** Mission progress function definitions for state space presented in Figure 4. All actions defined by reaction rules are assumed to take 1 unit of time. It is worth noting that $f_{3}$ function requires both UAVs to be in the same time (variable *z*) in order to return something other than 0.

Function Identifier	Function Definition
$f_{1}$	$f_{1}\left(c,t\right)=\left\{\begin{array}{cc} \left[\langle (b,y),(a,x+1)\rangle ,\Omega \cup \left\{r1_{t+1}\right\}\right] & :\;c=\left[\langle (a,x),(b,y)\rangle ,\Omega \right]\\ 0 & :\;c=0 \end{array}\right.$
$f_{2}$	$f_{2}\left(c,t\right)=\left\{\begin{array}{cc} \left[\langle (a,x),(b,y+1)\rangle ,\Omega \cup \left\{r1_{t+1}\right\}\right] & :\;c=\left[\langle (a,x),(b,y)\rangle ,\Omega \right]\\ 0 & :\;c=0 \end{array}\right.$
$f_{3}$	$f_{3}\left(c,t\right)=\left\{\begin{array}{cc} \left[\langle (a,z+1),(b,z+1)\rangle ,\Omega \cup \left\{r2_{t+1}\right\}\right] & :\;c=\left[\langle (a,z),(b,z)\rangle ,\Omega \right]\\ 0 & :\;c\neq \left[\langle (a,z),(b,z)\rangle ,\Omega \right] \end{array}\right.$
$f_{4}$	$f_{4}\left(c,t\right)=\left\{\begin{array}{cc} \left[\langle (b,y),(a,x+1)\rangle ,\Omega \cup \left\{r1_{t+1}\right\}\right] & :\;c=\left[\langle (a,x),(b,y)\rangle ,\Omega \right]\\ 0 & :\;c=0 \end{array}\right.$

**Table 3 sensors-21-00622-t003:** Mapping of labels in the state space to reaction rules in the TTS for scenario 1.

Label of Change	Reaction Rule
$m1^{i}$	*uta mov atx* occurring in *i*th transition
$m2^{i}$	*utb mov atx* occurring in *i*th transition
$m3^{i}$	*mov atx2y* occurring in *i*th transition

**Table 4 sensors-21-00622-t004:** A schedule of actions for both UAVs based on the walk of the form $0\overset{m1_{1}^{1}}{\to }1\overset{m1_{2}^{5}}{\to }0\overset{m1_{3}^{1}}{\to }1\overset{m2_{4}^{4}}{\to }4\overset{m3_{5}^{11}}{\to }5$. Each (x,y) element denotes: *x*—last scheduled action, *y*—the time moment since the *x* action is performed.

	START	$m1_{1}^{1}$	$m1_{2}^{5}$	$m1_{3}^{1}$	$m2_{4}^{4}$	$m3_{5}^{11}$	END
$UAV_{A}$	(−,0)	(m1,0)	(m1,1)	(m1,2)	(m1,2)	(m3,3)	(m3,3)
$UAV_{B}$	(−,0)	(−,0)	(−,0)	(−,0)	(m2,0)	(m3,3)	(m3,3)

**Table 5 sensors-21-00622-t005:** A list of controls with their respective interpretations.

Control	Interpretation
A	A map area
UAV	An active Unmanned Aerial Vehicle
DUAV	An inactive Unmanned Aerial Vehicle
B	The base
DS	A source of information
DU	A unit (piece) of information

**Table 6 sensors-21-00622-t006:** A mapping of system changes labels to corresponding reaction rules.

Label of A Change in A System	Corresponding Reaction Rule
mov	move
mib	move into base
mob	move out of base
ded	deploy data
dod	download data
dea	deactivate uav

## Data Availability

The data presented in this study are available on request from the corresponding author.

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
