# Peer review of "UAV Swarms Behavior Modeling Using Tracking Bigraphical Reactive Systems"

_sensors, 2021, doi:10.3390/s21020622_

Round 1
Reviewer 1 Report
The paper deals with a method for modeling UAV swarm missions and determining behavior for the swarm elements. Whether the theoretical derivation in the paper is right, the paper itself does not match the scope of the journal. That is, the reviewer fails to figure out anything related to the sensor, rather the paper is all about the mission management of multiple UAVs in topological data structure (bigraph) in order to determine all possible behavior policies. This is For this reason, the paper might be more suitable for other journal, for instance, MDPi Aerospace journal, which covers the topics such as multidisciplinary design optimization, software engineering, data analysis, artificial intelligence, aerospace vehicles' operation, control and maintenance, to name a few.
Regarding the technical correctness of the paper, the reviewer finds that the paper is composed of two parts: the theoretical basis in Sec. 2 and two example cases in Sec. 3. First of all, the Section 2 is not clearly explained for general audience. Rather, the authors defines UAV swarm mission as Tracking Bigraphical Reactive System (TBRS) with using bigraph and bigraphical reaction rules in very complicated manner, followed by the definition of bigraph structure, bigraphical reaction rules using the aforementioned definition, and another definition of state space. The reviewer gets completely lost in understanding how each definition is correlated and how the definitions are applied in their method. For this reason, the reviewer is afraid to say that the authors fail to clearly explain why they choose such data structure (definition) and the benefit of using in their application.
The second part in Sec 3 describes two example cases, however as the reviewer's point of view, it seems like just two simple test cases of their modeling method (TBRS). In order to claim the advantage of the authors' proposed method to achieve of adaptation and cooperation in complex scenarios, the Section 3 should be rigorously improved.
Reviewer 2 Report
The abstract should contain a clear statement about the significance/novelty of this research and a clearer depiction of the key conclusions arising from this work.
Currently, the abstract states: “The method was verified on two scenarios and two metrics were proposed to measure a size of a modeled system. The key finding of the study is the algorithm for determining all possible behavior policies resulting in a specific state and consisting of a given number of actions.”
This text is clearly inadequate to describe the key contributions of the paper and does not reflect the key findings of the case studies. It should also be clarified that only simulation case studies were carried out.
The main issue that I have with this paper is that it is out of scope for MDPI Sensors. In particular, there is no “Sensors” contents at all in the paper and the presented methodology is intended for aerial robotics behaviour modelling, which is an interesting topic but clearly outside the scope of MDPI Sensors.
I recommend that this paper be withdrawn/rejected and resubmitted to MDPI Robotics (Aerospace Robotics and Autonomous Systems Section), where it will likely encounter a more specialised audience and, consequently, receive more reads and citations.
Reviewer 3 Report
The paper presents an original method to model swarm missions and plan the behaviors of robots. This method is based on bigraphs that can integrate the different tasks and agents involved in the mission. Two cases of use of the method are presented to demonstrate its work.
In my opinion, the proposed method is promising and can be very useful for researchers on swarm robotics. Therefore, I consider that paper can be accepted after some minor changes:
- The description of the method is sometimes hard to understand. Then, the examples really help the reader to understand how it works. Please consider reinforcing the description of the method, complementing the mathematical description with sentences that remember the information of previous steps, or pose an example of the present one.
- As you describe in the introduction of your paper, swarm robotics is a wide world. You place your method in this world as bottom-up, non-adaptive, automatic, and macroscopic. Then, you show two examples of application to go to a target area and collect information from different sources placed in different areas. However, I miss some discussion about the applicability of the method. In which missions can your method be applied? In which missions is it better than other methods?
- In the same way, I miss some comparisons between your method and its main alternatives. For instance, some works of behavior-based algorithms propose combining different behaviors that can be optimized to adapt to different scenarios. Which are in your opinion the advantages and disadvantages of your algorithm?
- You state in line 42 that "Currently, there is no commonly accepted definition of robots swarm, all of them however link robotic swarms to multirobot systems", but later you introduce the most common definition of robot swarm. I suggest you rewrite or remove this sentence.
- There are language and typo mistakes. Some of them are the following:
- robots swarm -> robot swarm (multiple times)
- If a swarm is heterogeneous it consists of multiple heterogeneous subgroups -> I think the second word should be homogeneous (60)
- nonadaptive -> non-adaptive (multiple times)
- This is done by finding a walk in the state space between a vertices representing an initial and (possibly, one of many) final state (being a desirable outcome of mission). -> I find this sentence confusing. Please, rewrite it (213)
- Place graph is intended to model a spatial relations between system’s elements. -> Articles are wrong: Place graph is intented to model spatial relations between the system's elements (220). I suggest you review this kind of issue in the whole paper.
Round 2
Reviewer 1 Report
The manuscript is properly revised in accordance with the reviewer's comments.
This manuscript is a resubmission of an earlier submission. The following is a list of the peer review reports and author responses from that submission.